# ML Reproducibility Systems: Status and Research Agenda

## Abstract

As companies are more and more leveraging the power of machine learning (ML), ML models, the data they were trained on, and the training pipelines themselves, are becoming increasingly important assets. Hence, a new set of tools is being developed, which aim to help manage the *ML model lifecycle* in a more structured way. As the model lifecycle involves a lot of experimental back-and-forth, reproducibility is an important aspect that such tools need to provide. However, as model lifecycle management is an emerging field, only few best practices on how to support reproducibility exist. This led to a large variety of tools being developed that all have the same goal of providing reproducibility but with major differences in how this goal is achieved. As a result, users are overwhelmed with having to navigate this vast tooling landscape and having to choose a tool that best fits their needs. This is a difficult task as the reproducibility capabilities of different tools can vary and users need to determine themselves, what is supported and what functionalities would fit their specific use case.

In this paper, our goal is to add structure to the process of deciding on a specific tool in terms of its reproducibility capabilities. We identify the most significant artifacts of the ML model lifecycle and, based on these, propose a generic classification framework that allows to assess the reproducibility capabilities of a specific tool and make it comparable to other tools. To evaluate our framework, we conduct an analysis of 12 popular ML lifecycle management tools. We study each tool in detail and classify it according to our framework. We then compare the tools to each other to determine what degree of reproducibility is provided in general and where reproducibility support is still lacking. While overall we find that the majority of tools offers most features that are required for full reproducibility, there still exist gaps in terms of automating reproducibility and cross-tool reproducibility. Based on these findings, we provide a set of research challenges which need to be addressed in order to better understand reproducibility and to fill the gaps of current systems.

## 1  Introduction

As the use of machine learning (ML) is becoming ubiquitous across industries, managing ML models is increasingly important. Similar to code, ML models are important assets that need to be developed, tested, deployed, and updated. However, this *model lifecycle* significantly differs from a traditional software development lifecycle, due to three main reasons: 1) The lifecycle itself is a complex workflow, consisting of a variety of phases including data preparation, model training, and inference; 2) ML model creation is *experimentation-based*, making it an ad-hoc process with frequent jumps between the different lifecycle phases [50]; 3) the different phases are connected through a tight feedback loop, which requires to rapidly incorporate new data into deployed models [45]. As a result, existing software engineering methodologies and best practices cannot be directly applied.

Over the past years, a large number of new lifecycle management systems have been developed to simplify the end-to-end process [1–4, 7, 8]. The aim of these systems is to structure, orchestrate, and automate the model lifecycle while integrating with other systems for the individual lifecycle phases, e.g., systems for model training [16, 20, 29, 34, 37], data cleaning and preparation [42, 47], parameter exploration and tuning [18, 32, 51], or model explainability [33, 43].

One of the core focus areas of lifecycle management systems is *reproducibility*. Due to the complexity of the lifecycle and various regulatory requirements, being able to reproduce a machine learning model and its predictions in different environments is a key challenge for successfully applying ML technology in production [50]. This challenge is amplified as ML reproducibility not only requires to be able to exactly reproduce a certain model but also to *track* the exact steps and inputs that went into the model to explain, compare, and reason about different models and their outputs.

To prevent an impending reproducibility crisis [23, 25, 49], both research and industry have spent significant effort on trying to incorporate reproducibility as a first-class citizen into the lifecycle management systems. The outcome is a large variety of different tools and platforms to support reproducibility. As a result, users find themselves in the situation of having to choose one or several systems to meet the requirements of their particular use case.

However, choosing the right system is not straightforward [17]. While on the surface, each system promises reproducibility and tracking, the actual capabilities (and their implementations) of different systems vary. For example, some systems provide tracking capabilities for code and models [2, 7], some support version control for input data [1, 14], and some focus on visualizing input parameters and output

metrics [15]. The different meanings and understandings of reproducibility add further confusion for users [39,46]. Based on these issues, we argue for a comprehensive framework, which allows to categorize and classify the reproducibility capabilities of different systems.

While previous work has aimed to formalize the ideas of reproducibility for machine learning and provide best practices [22,28,46]. these attempts did either not look at system capabilities directly and only provide a higher-level classification of reproducibility, or did not consider the specific technical capabilities a system needs to provide to support seamless and automated reproducibility of complex ML pipelines. Other work has empirically assessed the state of reproducibility for current machine learning research [23,31,40] but the main focus was on how reproducible work is rather than how reproducibility is achieved.

In this work, we attempt to add structure to the reproducibility capabilities of the vast ML lifecycle management system landscape. Our goals is to allows users to better define their reproducibility needs and pick the appropriate system (or combination of systems) for the task. Therefore, we devise a framework for assessing the reproducibility capabilities of a certain system.

The key idea of the framework is that artifact *versioning* plus versioning *automation* is sufficient to classify the reproducibility capabilities of different system. In the context of ML, artifacts refer to anything that is an input to or an output of the model lifecycle, e.g., the raw input data, training parameters, or the trained model. Versioning refers to the ability of keeping track of the different artifact versions that were used throughout the model lifecycle. Automation describes, how much user effort is required to create and retrieve specific versions of specific artifacts. Overall, this allows to classify reproducibility ranging from "no versioning, no automation", (no reproducibility) to "everything is versioned and automated" (full reproducibility).

The framework then consists of two main parts: 1) a list of artifacts, such as the training data or the input parameters to the training algorithm, which are necessary to successfully reproduce a model; 2) a two-dimensional classification scheme to assess, if and how each type of artifact is versioned in a certain tool and how much automation the tool provides to manage versions.

We test the applicability of our framework, by conducting a large study of the reproducibility mechanisms of 12 different ML lifecycle management systems. Each of these systems advertises tracking and reproducibility support for models. The goal of our study is to 1) provide an overview of the current state-of-the-art in ML tooling for reproducibility; and 2) use our framework to structure the reproducibility capabilities of the different tools.

Overall, we find that reproducibility support is good among ML lifecycle management systems in terms of versioning. Additionally, most systems also provide a good set of au-

tomation features to ease the reproducibility of experiments though feature-complete automation is rare. However, while reproducibility is well supported *within* the bounds of a specific system, reproducibility across systems is still hard to achieve. This is partly due to the fact that different systems model ML projects and their corresponding reproducibility in different ways without a common standard, which leads to incompatibilities and diverging feature implementations.

The results of our analysis lead us to propose a research agenda towards better understanding and providing reproducibility support in modern ML systems. The proposed research items include research problems that cover directions on how to extend our presented framework and study to gain deeper insights into reproducibility features and how to extend systems to improve their existing reproducibility features further.

After introducing the necessary background on the ML model lifecycle (§2), we make the following contributions:

- We introduce a comprehensive framework to classify the reproducibility capabilities of ML lifecycle management systems (§3).

- We use the framework to survey 12 different systems in terms of what reproducibility they provide and how they are providing it (§4).

- We present a research agenda towards a better understanding of ML reproducibility and the requirements on the corresponding systems (§5).

We discuss related work in §6 and conclude in §7.

## 2 Background

In this section we provide the necessary background on the individual phases of the model lifecycle and explain, why reproducibility is important in each of the phases (§2.1). We then describe the different types of reproducibility in order to clarify the terminology (§2.2) and to frame their distinct properties in the context of ML projects.

### 2.1 The ML Model Lifecycle

The lifecycle of an ML model comprises three stages: Experimentation, Development, and Operation.Each stage consists of many individual steps and represents an iterative process, i.e. individual steps are groups of steps and are often repeatedly run to explore different variations of a model. The three main stages also have feedback loops between each other.

**Experimentation.** This is the initial stage where a model is created. It involves steps such as data wrangling, feature engineering, exploratory model building, and finally selection of a "winner" model [48]. The last step in this stage is to

prepare the model so that it can be easily consumed by the following stage.

During experimentation, reproducibility is of high importance due to the iterative and exploratory nature of the stage [35, 50]. As data scientists repeat individual steps and try new algorithms, parameters, and data, it is essential to keep track of the different configurations in order to analyze the differences between different models and to be able to exactly recreate models that performed well [50].

**Development.** Once a model has been built, the next step is to consume it. This means evaluating the suitability of the model as part of a data-driven application (e.g. a predictive service, a forecasting chart, etc.).This involves first scoring the model to new scenarios (new datasets) and can be as simple as obtaining the model (e.g. downloading it from a model registry) and running it in the context of the application.

Reproducibility during this stage is required in order to, first, connect the model from the experimentation stage to the development stage and second, to keep track of the development environment.If a high-scoring model breaks or performs badly in the context of an application, it is important to understand the difference between the two environments and to identify the input that led the model to produce inaccurate results.

**Operation.** During the operation stage, the model and its application are deployed to production. Similar to traditional applications, this might involve testing on multiple environments (dev, staging, and production).Additionally, this stage also involves monitoring of the model [19, 48] to find concept drift, performance issues, staleness, etc.

It is important to have reproducibility support during operation as application operators are often required to explain the performance of a production model [44]. Explanations are necessary in order to be able to debug production issues but can also be a dictated by different regulatory requirements. For explanations to be useful, it is crucial that individual predictions of a model can be reproduced and traced back all the way to the experimentation stage.

In this work, our focus is on reproducibility during the experimentation stage due to the fact that most tools support (and focus on) this stage. We outline future research directions on other lifecycle phases in §5.

## 2.2 Reproducibility, Repeatability, and Replicability in ML

Reproducibility as an umbrella term is often split into three different, more precise concepts: Reproducibility, Repeatability, and Replicability [21, 39]. The ACM definition of each of these three terms, is the following (as illustrated in Figure 1) [6]:

- **Repeatability.** "The measurement can be obtained with stated precision by the same team using the same mea-

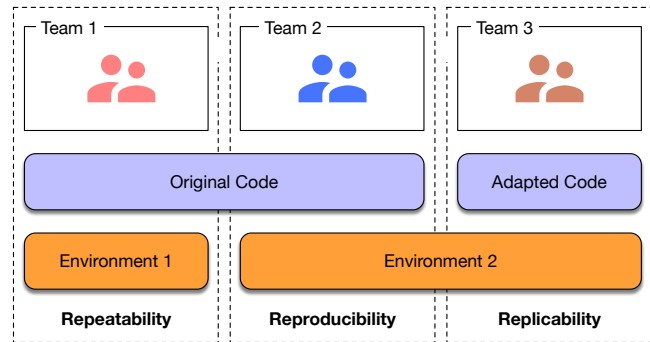

Figure 1: The different types of reproducibility

surement procedure, the same measuring system, under the same operating conditions, in the same location on multiple trials. For computational experiments, this means that a researcher can reliably repeat her own computation." *This refers to reproducibility by the same team with the same experimental setup.*

- **Reproducibility.** "The measurement can be obtained with stated precision by a different team using the same measurement procedure, the same measuring system, under the same operating conditions, in the same or a different location on multiple trials. For computational experiments, this means that an independent group can obtain the same result using the author's own artifacts." *This refers to reproducibility by a different team with the same experimental setup.*

- **Replicability.** "The measurement can be obtained with stated precision by a different team, a different measuring system, in a different location on multiple trials. For computational experiments, this means that an independent group can obtain the same result using artifacts which they develop completely independently." *This refers to reproducibility by a different team with a different experimental setup.*

The above definitions require to clarify three different concepts in the context of the ML model lifecycle: 1) the artifacts; 2) the experimental setup; and 3) the precise meaning of "same".

**Artifacts.** From the above ACM language, we can equate "artifacts" to the distinct components of an ML project. These include components such as the input data, the algorithm and its parameters, the software and hardware environment, and the outputs such as the ML model (we will introduce a comprehensive set of artifacts as part of the framework description in §3.1).

**Experimental Setup.** The setup is defined differently depending on the stage of the lifecycle, as the output of one stage (models, model-related and general metrics) becomes the input of the next one. If we consider models/metrics as input

data for the development and production stages then in all stages there is code, data (raw/models/metrics), parameters, environment and hardware.

**Meaning of "same".** The word "same" is used above to qualify both *setup* and *results*. Given the imprecision of the word itself, it can be a source of subjectivity and needs to be specified further. In the case of *results*, it can be defined as "a statistically sound comparison of original vs. new output" (as employed by previous studies [41]) and also depends on the stage of the lifecycle. For the experimentation stage, this means obtaining models that are as performant as the original ones. For the development and operation stages it means obtaining performance metrics that are as good as the ones obtained with the original data.

In the case of *setup*, we can objectively define equality by considering any change in the version of a component as the criteria for considering it different. For example, if two executions of the same code are carried out on the same setup, the only difference being a change in the firmware version of the GPU being used, then this is considered replication.

Thus, we have the following definitions in the context of the ML model lifecycle:

- **ML Repeatability**. Obtaining the same results by the same person using the same data, code, and hardware/software environment. Here, results are model and metrics for the experimentation stage and the performance of the model on the new data for development and operation stages.

- **ML Reproducibility**. Same as above but by someone else, possibly using equally spec'd hardware (e.g., same machine types, but not necessarily the exact same physical machine).

- **ML Replicability**. Same as above but by someone else and with changes to any of data, code, hardware/software environment.

For ML projects, the most important properties are repeatability and reproducibility. While replicability is relevant, the sensitivity of different models to their input and the environment makes replicability harder to achieve (see also §5).

## 3 Classification Framework

We now introduce our framework for classifying ML systems according to their reproducibility support. The framework consists of two core parts: 1) the project *artifacts*, i.e. the components of an ML project that are needed for reproducing an output such as a model (3.1); and 2) the reproducibility *capabilities* (§3.2) that are required for a system to provide reproducibility.

### 3.1 ML Project Artifacts

The project artifacts determine the set of components of an ML project that are decisive for the output of the project and hence, essential for achieving reproducibility. We identify 9 core components of an ML project: Code, input data, input parameters, output data, output metrics, software dependencies, system dependencies, hardware dependencies, and the overall experiment.

**Code.** At each stage of the entire model lifecycle, code is at the center of the practice. During experimentation, the focus of our work, code describes how data is cleaned and prepared, and which algorithms to use for model training.

**Input data.** Input data describes the raw data that is wrangled in order to produce the training, test and validation sets required to build a model during the experimentation stage. The raw data can be updated as new and more data becomes available.

**Input parameters.** In most cases, code and the corresponding data processing and training algorithms are heavily parameterized, including information such as a model's hyperparameters or the thresholds of a filtering algorithm. Parameters evolve separately from code and thus are treated as a separate component.

**Output data.** The output data of a project is any asset that is produced by a lifecycle stage and that is required by the next stage (or to provide functionality during the operation stage). For the experimentation stage, output data is usually a trained model.

**Output metrics.** Output metrics are relevant measurements captured during the different stages of the lifecycle. For the experimentation stage, metrics usually describe, how model properties change over time. Metrics include scores such as accuracy, recall, etc.

**Software dependencies.** These type of dependencies describe the direct dependencies that are required for a project to run. If these dependencies are not present, executing a lifecycle stage will fail. Such dependencies include necessary software packages, e.g., for python or R.

**System dependencies.** This type of dependencies describes system-related information associated to the execution of a lifecycle stage. While these are not explicitly defined as dependencies by a project, they can still influence the outcome. Examples include the specific operating system, software runtime versions, or the version of the kernel.

**Hardware dependencies.** These dependencies describe the hardware platform on which the code was executed, including information such as the specific type of GPU that was used for training. As the specific hardware can potentially influence the output of a project [36], it needs to be treated as a separate component.

**Experiment.** The experiment describes an entire lifecycle

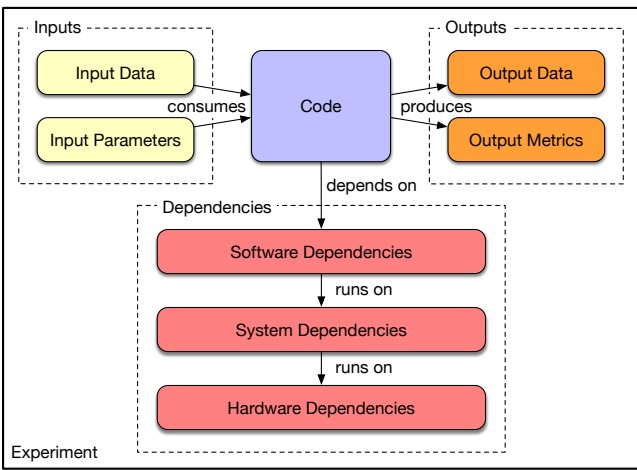

Figure 2: Interactions between project artifacts

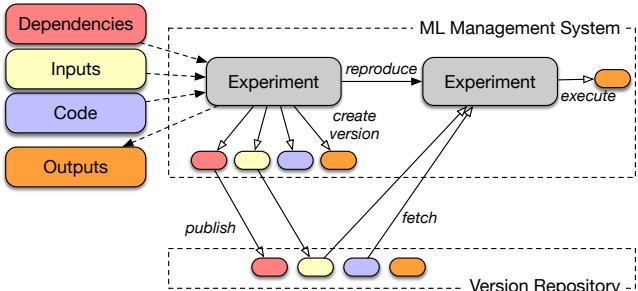

Figure 3: Versioning and automation for reproducibility

2. Apply the same processing steps to the inputs using the same algorithm and parameter/system configurations;

3. Compare the result to the previous result to verify equality of the reproduced output.

While the exact set of inputs can vary for Replicability, the overall workflow remains the same.

At the core of the above workflow is the ability to retrieve historic data. During the first step, the input data from the time of the original model creation needs to be retrieved. In the second step, the algorithms, parameters, and system configurations, which were used to create the original model, need to be retrieved. Finally, the original outputs need to be obtained to be able to compare the reproduced model to the original one. To be able to retrieve historic information, a system needs to track the individual versions of this information. Hence, *versioning* capabilities are a necessary requirement for any reproducibility system.

Code versioning is a standard practice when writing software in order to keep track of how code has changed over time. However, for ML projects, versioning is more complicated as not only code but all other relevant artifacts needs to be tracked. Whereas code versioning is well understood and has a rich and mature set of tools [5, 9, 12], versioning of ML project components such as input data or software dependencies is still in its early steps.

**Versioning Actions.** In our framework, versioning has three corresponding actions, which describe if and how versioning is supported: None, snapshot-based, and version-based. *None* means that versioning is not supported.

*Snapshot-based* means that the versions of the different artifacts for a *single* experiment are kept as part of the experiment execution. For example, a snapshot of the input parameters is taken before the start of the experiment, stored in a database, and linked to the specific experiment. This snapshot can then later be retrieved as part of the past experiment to determine the exact parameter configuration. Snapshots, however, do not document relationships *between* experiments, i.e. they do not keep track of, how and why, parameters changed *over time*.

phase as a whole and is a combination of all the artifacts as described above. While this may seem redundant, it is important to treat the experiment as a separate artifact as it can provide additional context for what the overall aim of a particular combination of artifacts was, e.g., train a deep learning model for image classification based on the ImageNet dataset on a cluster of 10 GPUs.

Figure 2 shows how the different artifacts relate to each other. Code is at the core, consuming both input data and input parameters and producing output data and output metrics. To successfully run, the code requires a set of software dependencies, which in turn, run on a system with a certain configuration. At the bottom is the hardware on which the experiment is run.

To achieve perfect reproducibility of an ML model, information about the state of all the 9 artifacts, at the time when the original model was created, needs to be available to users. Any one artifact can have an influence on the expected outcome, i.e. if an artifact is not kept track of properly, reproducibility may be affected.

## 3.2   Reproducibility Capabilities

Given the artifacts that are required for reproducibility, the next part of our framework defines the capabilities that are required to utilize those artifacts for reproducibility. Our framework is built around two core capabilities: Versioning and Automation. Each of these capabilities has associated *actions* that describe what exactly a system supports along those two dimensions.

**Versioning.** Looking at the types of reproducibility as described in §2.2, we can identify a common workflow that is required for reproducing any output:

1. Obtain the inputs from the point in time for which the output should be reproduced;

*Version-based* means that the project artifacts are kept in full version control. Temporal changes are kept track of through commits and the evolution of those changes is documented through the commit history and the corresponding commit messages. Individual commits for different artifacts are linked to experiment executions. Additionally, full version control supports branching, merging, and reverting changes back to a previous commit. This approach is equal to how code is versioned in the majority of software projects.

**Automation.** In theory, versioning of each project artifact is enough to achieve full reproducibility. If the versions (or snapshots) of every artifact of an experiment are available, a data scientist will be able to rerun the experiment and compare the results to the original execution. However, manually creating the correct experimental setup and retrieving the correct versions associated with the target experiment is often difficult, in particular for complex ML workflows. Hence, our framework considers automation capabilities as an essential part for providing reproducibility.

Automation refers not only to the ability to reproduce an experiment but also the ability to *create* a reproducible experiment. It aims to classify a system according to two main questions:

1. How easy is it for a data scientist to replicate a previous experimental setup, rerun a certain experiment, and compare the output to the original results?

2. How easy is it for a data scientist to create all the necessary artifact versions, capture the experimental setup, and create a bundle that allows to reproduce an experiment?

A perfect reproducibility system makes it easy to both create artifact versions and provide them as a reproducible bundle and to consume that bundle to rerun and verify past experiments.

**Automation Actions.** Our framework defines four main actions to describe automation capabilities: None, creation, publishing, and fetching of an artifact version. For code artifacts, an additional "execute" action is defined. Again, "none" means that no automation capabilities for a particular artifact are supported. In contrast to the versioning dimension, the automation actions are not mutually exclusive. It is possible for systems to support either none, or a combination of 1, 2, or all 3 (4 in the case of code) actions for an artifact.

*Creation* refers to the ability to automatically create artifact versions for an experiment. Based on the system's and artifact's versioning support, this could either mean that creating a snapshot or committing a new version of an artifact is automated. This action is part of a system's ability to create a reproducible experiment.

*Publishing* describes, whether a system allows data scientists to make artifact versions available so that they can be shared with others. Our framework does not consider whether

Table 1: The classification framework

| Artifact | Versioning | Automation |
|---|---|---|
| Code | N or S or V | N or (C and/or P and/or F and/or E) |
| Input Data | N or S or V | N or (C and/or P and/or F) |
| Input Parameters | N or S or V | N or (C and/or P and/or F) |
| Output Data | … | … |
| Output Metrics | … | … |
| Software Deps. | … | … |
| System Deps. | … | … |
| Hardware Deps. | … | … |
| Experiment | … | … |

artifacts are available publicly or require certain access credentials as long as they can be shared through some means that do not require to manually copy and transfer them. Again, this action is part of supporting the creation of reproducible experiments.

*Fetching* is the ability of a system to automatically retrieve previously published artifacts and use them when reproducing an experiment. If fetching is supported for an artifact, data scientists do not have to deal with picking and retrieving the right version themselves. This action is part of consuming a reproducible experiment.

*Execution* is a special action, only referring to code artifacts. As the actual experiment workflow is specified in code, code not only needs to be fetched but also executed in order to reproduce an experiment. All other artifacts do not require to (and cannot) be executed. Again, this action is also required to support consumable reproducible experiments.

**Framework Summary.** Combining the artifacts and the reproducibility capabilities yields the classification framework as summarized in Table 1. In the versioning dimension, N stands for None, whereas S and V describe snapshot-based and versioning-based version support, respectively. Along the automation dimension, C, P, F, and E are creation, publishing, fetching, and execution of artifacts, respectively.

## 4 System Evaluation

We now present our evaluation of the different systems. We first introduce how we generated the list of systems to evaluate (§4.1) and then use our framework to classify the systems according to their reproducibility capabilities (§4.2).

### 4.1 The Systems

Compiling a list of systems for analysis is challenging due to the large number of existing machine learning systems. To create our list of systems for classification and make it as complete as possible, we follow the methodology as described below.

Table 2: List of evaluated tools

| Name | Website | Version |
|---|---|---|
| ClearML | https://clear.ml/ | 17.04 |
| Dessa Atlas | https://www.atlas.dessa.com/ | 0.1.1 |
| Determined AI | https://determined.ai/ | 0.14.3 |
| Dolt | https://www.dolthub.com/ | 0.24.2 |
| DVC | https://dvc.org/ | 2.0.13 |
| Kubeflow | https://www.kubeflow.org/ | 1.3.0 |
| MLflow | https://mlflow.org/ | 1.17.0 |
| Pachyderm | https://www.pachyderm.com/ | 1.11.7 |
| Polyaxon | https://polyaxon.com/ | 1.9.5 |
| Qri | https://qri.io/ | 0.9.13 |
| Quilt | https://open.quiltdata.com/ | 3.4.0 |
| TensorBoard | https://www.tensorflow.org/tensorboard | 2.5.0 |

As a starting point we used a recently published list of over 200 machine learning systems, which aims to cover the entire ML tooling landscape [26]. As the list is still being updated, we took a snapshot of the list on 08/21/20 to fix the base set of systems. The snapshotted list consists of 212 systems in total.

As the base list covers *any* system related to machine learning, we needed to filter the list to extract only systems that offer some form of reproducibility support. For that, we manually looked at the website and documentation for each individual system and searched for mentions of keywords that indicate, whether the system provides some form of reproducibility. If capabilities such as "tracking", "reproducibility", "provenance", "versioning", etc. were mentioned, we included the tool in the list of reproducibility tools.

The initial filtering step reduced the list of systems to 43. To get a list of usable systems, we further filtered the 43 systems, removing any systems that are proprietary, inactive, only provide hosted solutions, or mention reproducibility as a planned feature in the future. This resulted in the final list of 12 systems as summarized in Table 2. We make the complete list of systems with filtered systems and the reason for filtering highlighted available as part of the supplemental materials.

## 4.2 Framework-based Classification

Next, we study the different tools by classifying their capabilities using our framework. To identify the versioning and automation capabilities for each artifact, we use information available through the system's official documentation. However, the documentation alone is often not sufficient to determine all relevant operational details of a system. Hence, in addition, we also create single-node test deployments for the individual system and observe them in action, specifically trying to trigger the different reproducibility capabilities for the different types of artifacts.

The results are summarized in Table 3 and our full analysis, with the reasoning behind the individual classifications is also made available as part of the supplemental materials. Table 3, each system has two rows, the first row for its versioning capabilities and the second row for its automation support. Based on the results, we make 7 key observations.

**Observation 1: Three types of systems.** The first observation is that there are three main types of systems: project management systems (ClearML, Dessa Atlas, Determined AI, Kubeflow, MLflow, Pachyderm, Polyaxon), data versioning systems (Dolt, DVC, Qri, Quilt), and visualization systems (TensorBoard). Each class of systems has a different objective, which is reflected in their capabilities.

The aim of *project management systems* is to provide ways for organizing and managing the entire or significant parts of the model lifecycle. Hence their capabilities are broad and mostly cover all dimensions. *Data versioning systems* focus on versioning input data. As input data can be large, versioning it is difficult and requires specific methods to achieve versioning in an efficient way. As can be seen in Table 3, the capabilities of data versioning systems are mostly placed around input and output data. However, some provide additional capabilities to support experiment management. The main goal of *visualization systems* is to provide a flexible way of tracking and visualizing experiment input parameters and output metrics.

The above observation means that to achieve full reproducibility for ML projects, the use of a project management system is required. If stronger forms of data versioning and more flexible ways of visualizing results is needed, project management systems can be paired with data versioning and visualization systems.

**Observation 2: Full versioning is rare.** When looking at the system capabilities, we can see that snapshotting is the dominant form of keeping track of project artifacts whereas full versioning is rare. Instead of systems allowing to keep track of individual artifacts with full version control (commit, branch, merge, etc.), they mostly store the current value of an artifact at the time of an experiment as a snapshot.

We believe that the main reason for this is that for smaller data, the majority of tools assume that it is already kept in a version control systems (mostly `git`) so separate version control capabilities would be redundant. Additionally, snapshotting is sufficient to keep track of the artifacts of an experiment while being much simpler to implement than fully-fledged version control. On the other hand, versioning large data is a less well understood problem and as data is becoming as (or even more) valuable as code, new solutions are created to offer data versioning support.

To achieve reproducibility, snapshotting is sufficient. If the state of each artifact at the time of an experiment is known, the experiment can be reproduced. However, as we will discuss in §5, versioning can be useful to enable other capabilities,

Table 3: Framework-based reproducibility tool classification

| Name | Code | Input Data | Input Params | Output Data | Output Metrics | Software Deps | System Deps | Hardware Deps | Experiment |
|---|---|---|---|---|---|---|---|---|---|
| ClearML | S C/P/E | S C/P/F | S C/P/F | S C/P/F | S C/P/F | S C/P/F | S P/F | N N | S C/P/F |
| Dessa Atlas | S C/P/F/E | S C/P/F | S C/P/F | S C/P/F | S C/P/F | S P/F | S P/F | N N | S C/P/F |
| Determined AI | S C/P/F/E | S C/P/F | S C/P/F | S C/P/F | S C/P/F | S P/F | S P/F | N N | S C/P/F |
| Dolt | N N | V C/P/F | N N | V C/P/F | N N | N N | N N | N N | N N |
| DVC | S C/P/F | V C/P/F | S C/P/F | V C/P/F | S C/P/F | N N | N N | N N | V C/P/F |
| Kubeflow | S C/P/E | S C/P/F | S C/P | S C/P/F | S P/F | S P/F | S P/F | N N | S C/P/F |
| MLflow | S C/P/F/E | S C/P/F | S C/P/F | S C/P/F | S C/P/F | S F | S F | N N | S C/P/F |
| Pachyderm | S C/P/F | V C/P/F | S C/P/F | V C/P/F | N N | S P/F | S P/F | N N | S C/P |
| Polyaxon | S C/P/E | S C/P/F | S C/P/F | S C/P/F | S C/P/F | S P/F | S P/F | N N | S C/P/F |
| Qri | S C/P/F/E | S C/P/F | N N | S C/P/F | N N | N N | N N | N N | N N |
| Quilt | N N | S C/P/F | N N | S C/P/F | N N | N N | N N | N N | N N |
| TensorBoard | N N | N N | S C/P/F | N N | S C/P/F | N N | N N | N N | N N |

e.g., , reasoning about why a certain change was made to an artifact.

**Observation 3: Almost no support for automatic dependency discovery.** From Table 3 we can see that most project management systems provide snapshotting support for software and system dependencies. In most cases, this is done through running the pipeline steps as containers and then recording information on which container image was used for a particular step. Additionally, for software dependencies, several systems also provide other ways of snapshotting, e.g., , through conda environments or a `requirements.txt` file to capture python packages.

However, there is little to no support for automatically capturing any software/system dependencies. While systems capture information on dependencies (e.g. in the form of a container image), they do not capture the dependencies themselves. The only exception we found is ClearML, which records all python packages that were required for an experi-

ment run.

This does not affect the ability to reproduce an experiment as the information on which dependencies are required is still available. However, automating dependency discovery and assisting users in creating container images or other dependency collections could be useful to ease burden on users. This especially applies to system dependencies as software dependencies are often already managed by the users while system dependencies are "just there".

**Observation 4: No hardware dependency tracking.** The fourth observation is that none of the systems provide any capabilities to record hardware dependencies, i.e. on which platform the code was run or which specific devices (e.g., GPUs or TPUs) and device models were used. While some systems keep information on whether a GPU was required or, if it is a scheduled system, on which worker a pipeline task was executed, we did not find any capabilities to keep track of which specific hardware was used.

At first sight, tracking hardware dependencies seems less relevant for reproducibility. Data scientists are usually less concerned about on which platform their code will be running. However, hardware can influence the outcome of an ML experiment [36] and hence, tracking hardware dependencies is an important feature that can help to improve reproducibility. Such capabilities should be incorporated into existing systems.

**Observation 5: Limited system interoperability.** As mentioned in §3.1, our classification framework treats an ML experiment as a separate artifact, which can be versioned, published, and fetched. However, when trying to classify systems in terms of their experiment versioning and automation support, we realized that "experiment tracking" can cover a broad spectrum of capabilities, which can support different types of reproducibility.

One one end of the spectrum is the ability to create and execute a copy of an experiment in the same environment and using the same system through which the original experiment was run. This makes it very easy and convenient to reproduce an experiment but requires to have access to the original environment. On the other end of the spectrum is the ability to export experiments to a different environment, prepare the environment, and then rerun it. This is usually harder to achieve as environments can differ and experiments are often created packaged for execution through a specific system.

Due to this difficulty, most system we studied are on the former end of the spectrum in terms of experiment versioning and automation support. While some allow to download experiments as bundles or packages, others allow to clone or copy experiments for immediate re-execution (with the ability to edit parameters). If users have access to the system, reproducibility becomes easy. However, this also highlights an important problem, i.e. that of interoperability between systems. There is no easy way to copy an experiment to a different system and reproduce it there. Such a task would either require to manually retrieve all experiment artifacts and prepare the experiment for a different environment or set up the system. Both options are tedious and the latter might not even be possible. Hence, more work is necessary to allow more seamless interactions between the systems.

**Observation 6: Systems are similar but different.** As can be seen from Table 3, most systems within a class of systems support the same capabilities. For the way these capabilities are supported, we identified some common themes across the different systems. For example, managing software and system dependencies through container images is common or allowing to track input and output data through some form of artifact logging API is supported by several of the project management systems.

However, while systems show similarities in some dimensions, they can also differ significantly for other capabilities. For example, one obvious difference is the target environment of the (project management) systems. While some are targeted at more generic environments, systems like Kubeflow, Pachyderm, and Polyaxon are solely build for Cloud native, Kubernetes-based setups. Another difference is how systems manage model outputs. While some do not distinguish between model or other output data, other systems (e.g. ClearML, Determined AI, or MLflow) provide more functionalities to manage produced models. Other differences include input parameter tracking (various degrees of automated tracking support), the assumption whether git is present or not, and the way, experiments are abstracted (as the project directory, experiment checkpoints, or just at a logical level).

This observation shows that our framework is able to capture the essential capabilities for reproducibility. Despite the differences of the systems, our framework is able to abstract them to a comparable representation that shows their similarities at a higher level. It also confirms the previous observation that, due to the differences, system interoperability is difficult and hinders reproducibility across systems.

**Observation 7: Reproducibility support is good but classification is messy.** Overall, our findings show that project management systems have good reproducibility support as they provide snapshotting capabilities and automation around those for (almost) all project artifacts. Additionally, more specialized tools for data and metrics tracking can be combined with project management systems to enhance the base capabilities and provide more functionality to users. For example, Determined AI already relies on TensorBoard by default for visualizing output metrics.

On the other hand, we found that classification of the capabilities is difficult and even messy at times. This is due to the fact that there are currently no standardized definitions around what exactly an artifact comprises and what exactly a certain automation capability needs to provide. Hence, we often encountered ambiguities and needed to resolve them

using our best judgement (see "Definitions and Clarifications" in the supplemental material). We hope that this work can provide a starting point for discussion around what specifically constitutes reproducibility in the context of the ML model lifecycle and to develop clear concepts and definitions around the various requirements to provide full reproducibility.

## 5 Research Agenda

Our framework and system analysis is an initial step towards better understanding the system requirements to achieve reproducibility in the ML model lifecycle. However, it still leaves a plethora of open questions that need to be answered to achieve a truly reproducible lifecycle. In this section, we suggest possible research directions, which we deem important to build better reproducibility systems. First, we discuss, how our current system analysis needs to be deepened to gain a better understanding of the different system capabilities (§5.1). Then, we introduce extensions that need to be added to our framework to capture all aspects of ML reproducibility (§5.2). Finally, we present current gaps in the tooling landscape that need to be filled to improve reproducibility support (§5.3).

### 5.1 Deeper Tool Analysis

We propose three main research directions to gain a better understanding of the reproducibility capabilities of current systems: User Experience (UX), Performance, and Benchmarking.

**User Experience.** In this work, our system analysis was solely focused on whether and how a certain system supports a certain reproducibility capability, as defined by our framework. As shown in §4, often, many systems offer the same or a similar set of capabilities. However, these systems are still different as they made different design choices in terms of, e.g., how projects are specified, how artifacts are tracked, or how information is displayed. While these choices do not necessarily influence the capabilities, they can have a big impact on a user's experience when operating the system. Hence, an important question is: *How easy is it for a data scientist to create a reproducible experiment with a specific tool and how easy is it to reproduce this experiment?*

Answering this question requires to conduct a user study of the different systems. Such a study would consist of two parts: 1) define a specific ML task that needs to be achieved by a user. For example, this could be to build a classifier for some task with a certain accuracy target; 2) reproduce the resulting model. This should be done by the same user who created the initial model to evaluate the repeatbility, and by a different user to evaluate reproducibility. Measuring, how fast users can achieve those tasks will provide a better understanding of what design choices help to improve reproducibility.

Conducting such a study comes with several challenges. First, a task needs to be defined that is both specific, requires

a reasonably complex workflow, and stresses the system in terms of its experimentation capabilities by requiring many iterations. Additionally, the previous ML expertise of the participants can heavily influence the outcome, which needs to be taken into consideration. Finally, it is not clear how the replicability support of a system should be evaluated as part of such a study, i.e. how should the changes that are introduced to the experimentation process be defined. The outcome of the study can help to understand what features are more/less important for reproducibility and what features may still be missing.

**Performance.** Another aspect that we have not addressed in this work is the performance of the different systems. While a UX study helps to evaluate the features of a system, a performance study evaluates the implementation of these features. As shown in §4, the same features can be implemented in a variety of ways, which affects the performance and efficiency of a system. As data is becoming larger and pipelines are getting more and more complex, having performant, resource-efficient, and scalable implementations of the different features will become necessary to keep operating the systems. The research question in this case is: *How costly in terms of resources and processing overhead is it to support reproducibility in the ML model lifecycle?*

This questions requires a detailed performance analysis of the different ML reproducibility systems. The analysis needs to evaluate the different versioning and automation features of a system for each of the project artifacts. This includes a variety of tasks such as evaluating how long it takes to version/snapshot an artifact for different backend stores, the overhead this adds on the end-to-end pipeline, how much storage is required by a version/snapshot, how long it takes to retrieve past data, how long it takes to setup and rerun a past experiment, and many more.

The main challenge of such a study is to collect a representative set of workloads that is able to both stress each individual feature/artifact combination and evaluate the system as a whole. Additional difficulty is added as, similar to the UX case, not only the creation of reproducible artifacts but also the reproduction of those artifacts needs to be considered. The results will help to find, understand, and improve current bottlenecks in ML reproducibility systems.

**Benchmarking.** The above two questions naturally lead to the next research direction: Creating a benchmark for ML reproducibility. Questions of feature support and performance are traditionally answered by standardized benchmarks, which allow to compare different systems using a well-defined set of tasks. The results of the benchmark determine, which features are supported by a system and how efficiently the tasks can be executed. The same concept can be applied to ML reproducibility systems, which leads to the question: *How can a reproducibility benchmark be defined, in general and for ML reproducibility in particular?*

Creating a reproducibility benchmark requires to identify a set of representative workloads and corresponding parameters to configure the benchmark. This is similar to the above described UX and performance studies but goes a step further as it combines the two and requires to standardize the workloads. Hence, the outcome of the UX study and the performance study can be seen as necessary steps towards a reproducibility benchmark. Overall, having such a benchmark would tremendously help researchers and developers to better and more systematically understand the reproducibility capabilities of ML systems, and help users, to compare different solutions and pick the most suitable one for their needs.

## 5.2   Framework Extensions

The framework presented in this paper has several limitations as it does not consider a variety of aspects of the ML lifecycle. We propose two main extensions to the framework: Cover the entire lifecycle and consider scalability.

**Entire Lifecycle.** So far, we have focused on understanding reproducibility for the experimentation phase of the ML lifecycle. However, reproducibility is also required during the development and operation phases (§2.1). Hence, these phases need to be included in the framework to get a complete picture of how different systems support reproducibility end-to-end. The main question that needs to be answered in this case is: *What framework additions are required to assess its reproducibility capabilities for the entire ML lifecycle?*

The answer to this questions requires two main steps. First, any additional artifacts that are added by the development and operation phases need to be identified. For example, during operation, the incoming requests against the trained model would need to be tracked in order to reproduce the individual predictions the model made. Second, new automation capabilities that are introduced by the other phases need to be added to the framework such as automatic deployment of newly trained models or even online (re-)training of deployed models.

A major challenge for this extension is to gain a deeper understanding of the entire end-to-end lifecycle to clearly define, what new artifacts and automation needs to be added. This is difficult as the discipline of MLOps itself is still developing and best practices are still being identified and experience frequent changes. Hence, the framework extensions need to be flexible enough to allow for changes while still trying to capture the additions at a fundamental level.

**Scalability.** Another aspect not captured by the current framework is the scalability of the reproducibility capabilities. This refers to the ability of a system to reproduce artifacts that have been created as part of a distributed pipeline, e.g., where different pipeline steps where executed on different hosts and individual steps, such as training, were run in parallel on a cluster. While a system might be able to capture the pipeline in general, it may not be able to recreate the distributed envi-

ronment used for executing the pipeline. The research question in this case becomes: *What framework additions are required to assess the scalability of the reproducibility capabilities of a system?*

Answering this question again requires to understand two main aspects of the system: 1) How does the system enable a data scientist to create scalable pipelines; and 2) How does the system enable a user to reproduce these scalable pipelines. Creation likely will require to add new capabilities to the framework, e.g., how can users define pipelines and how is the execution of these pipelines automated on different environments. Reproduction also will require additions such as new artifacts for the cluster configuration/dependencies and if and how a distributed environment can be retrieved and set up automatically.

## 5.3   Missing Capabilities

We identify two main missing capabilities, that ML reproducibilty systems could benefit from: Reasoning and Sensitivity. Additionally, we also introduce the need for more standardization in managing ML porjects.

**Reasoning.** For all types of ML reproducibility, it is necessary to be able to rerun an experiment and compare its outcome to the original experiment. However, if something fails during the rerun or the outcomes are not identical, data scientists are often facing the difficult task of identifying what failed and why and why the outputs do not match. Reasoning capabilities help data scientists with this task by pointing to differences in the setup, providing explanations of the original experiment's intentions, and streamlining the comparison process. This direction poses two main questions: *What reasoning capabilities are required for reproducibility and how can they be achieved in a system?*

One example for systems that offer reasoning capabilities are notebooks such as Jupyter. Notebooks showcase the original intentions of the data scientist through comments and explanations, which are part of the pipeline execution. While this approach is useful to understand the individual steps, it does not provide guidance in case a step fails or does not produce the desired output. To support this, additional capabilities are likely required, e.g., capturing the provenance for each individual artifact and explanations of the individual artifact changes.

**Sensitivity.** Another step towards better reproducibility support in the field of ML (and in general) is a better understanding of what aspects matter the most to achieve accurate reproducibility. While our framework tries to capture the most important input artifacts, we do not yet understand, how important each of these artifacts is individually and how it influences reproducibility. A good example for this is hardware dependencies. As shown in §4, no framework provides a way of capturing hardware dependencies. This is because it is difficult to capture hardware dependencies but also, be-

cause the general understanding is that, e.g., input data is much more important for accurate reproducibility. While this seems intuitive, there is no systematic way of quantifying the influence of hardware or input data on the output and hence an important question is: *How sensitive are the reproduced outputs to the different input-related artifacts?*

Answering this question requires a sensitivity analysis. For each input artifact, a range of values needs to be defined and for each of those values, the output and the difference to the original output needs to be measured. This will offer an insight into which artifacts are most important and need to be precisely kept track off and which ones, if any, are less important and might not need to be tracked at all. We envision a tool, similar to `git bisect` [10], could be used for this task.

A major challenge for such a sensitivity analysis is how to define the value ranges for different artifacts. While some artifacts, e.g., hyperparameters, can be easily captured in ranges, it is less clear for others, e.g., software dependencies or input data. It becomes even trickier if a model output has a large number of software/system dependencies as even across those dependencies, some might be more or less important. While not straightforward, a sensitivity analysis can provide valuable insight into which artifacts require more attention to offer better, more accurate reproducibility.

**Standardization.** One of the main findings of our classification is that systems provide good reproducibility support as part of their own functionalities but *cross-system reproducibility* is still hard to achieve. This is due to the variety of ways in which systems define and manage projects. This can make it hard for a team to reproduce another team's results if they each choose a different management system to organize their projects. Having to deploy another system just to be able to reproduce someone else's results can be prohibitive and hinder reproducibility (and specifically replicability as defined in §2.2).

As a consequence, standards are needed in this area to facilitate the exchange of projects and allow seamless import/export from/to different system. While there is some effort in defining common conventions (e.g. through Mlflow projects [11] and MLflow models [13]) there is currently no universally accepted standard that would allow to easily exchange data between different system. The community needs to come together to drive this effort to ensure that in the future, sharing and reproducing different ML experiments becomes a commodity.

## 6 Related Work

Recent efforts have analyzed published articles in the field of AI and ML [23, 27, 31, 40], with the goal of assessing the reproducibility of articles published in this area. In general, this body of work analyzes a set of articles in order to extract a list of features for each paper, obtaining a dataset that is analyzed

and for which statistical descriptors are presented. Further analysis can be done, such as the one presented in [40], where correlations between these features for each article, such as how much readability is affected by availability of code, are analyzed. General guidelines associated to what ML platforms can provide to support reproducibility is outlined. Our analysis focuses on systems and experiments rather than article content, and does not go into identifying relationships between features, although we mention what follow up analysis could be done in this regard (see §5).

In addition to analyzing the state of the art by looking at published articles, existing work has surveyed existing ML tools with the goal of evaluating their reproducibility capabilities, mostly from the point of view of users. These have been both qualitative [30, 38] and quantitative [24, 28] evaluations. Specifically for the latter, existing work has proposed frameworks for systematically quantifying the reproducibility features of ML tools. The main difference between the present work and these previous efforts is our focus on quantifying the degree to which the two fundamental features involved in reproducibility (versioning and automation) are supported by these tools. In addition, we provide a research agenda towards a better understanding of reproducibility systems for ML and improving their utility for end users.

## 7 Conclusion

In this paper, we attempted to better understand the requirements of reproducibility for ML applications and assess the reproducibility support of existing ML lifecycle management systems. We proposed an evaluation framework based on the two core reproducibility principles of versioning and automation to classify the reproducibility capabilities of ML systems. Using this framework, we carried out an in-depth study of 12 popular ML systems.

Our findings reveal that most systems have comprehensive support for reproducibility but there are still gaps, such as missing hardware dependency tracking, automatic dependency discovery, and better cross-system reproducibility. We outlined a research agenda for filling these gaps and improving our understanding of ML reproducibility in general. We hope that the community can work together towards these goals and help make reproducibility for ML applications a seamless process for data scientists.

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
