# OpenReview forum: "ML Reproducibility Systems: Status and Research Agenda"
_JSYS/2021/Nov_Papers — Submitted to JSYS Nov 21_

### Official Review · Reviewer_nf5E · 2021-12-01
**A paper highlighting an important issue, but lacking details on the compared tools and presenting a research agenda ignoring existing research areas.**

**Decision:**

Weak reject: interesting papers with flaws, not sure if they can be fixed in three months

**Review:**

Disclaimer on expertise: While I am actively publishing in the area of continual/lifelong machine learning, I am not very familiar with state-of-the-art reproducibility systems and the tools presented in this paper.

# Summary

This paper presents a framework to compare management systems for machine learning (ML) code and data in terms of versioning and automation capabilities. These two aspects are critical to making ML systems more reliable to reproduce. The paper provides a solid overview of a typical ML workflow, ML artifacts, and differences between repeatability, reproducibility, and replicability and considers a large number of systems as candidates for their framework. In the end, it finds that 12 systems out of all candidates in some way aim to assist reproducibility and compares them as follows: whether they only store nothing, single snapshots, or full version histories for the different artifacts; and whether they allow automating creating/publishing/fetching/executing the artifacts. Finally, the paper presents ideas to continue the analysis of these tools as well as some points that could be part of a future research agenda for ML reproducibility.

## Strengths

- The paper highlights the important challenge of reproducibility of ML, and why this goes beyond just the code.
- The list of ML artifacts and the potential versioning/automation capabilities are comprehensive
- The paper points out the need for standardization.

## Weaknesses

- The actual analysis of tools is rather shallow, the paper only reports whether a tool checks a certain box or not, and does not provide much-needed detail into *how* a tool implements these capabilities, which in my opinion would be essential to decide between tools.
- The paper does not motivate the benefits of specialized tools for versioning and automation, in particular, what these tools provide compared to a "tried and prove" pipeline consisting of version control with git and continuous integration/deployment.
- The future research agenda is highly idealized and fails to fully convey the complexity of some of the raised tasks, e.g. how difficult it is to capture the sensitivity of ML models to input data.
- The paper lacks any discussion of the limitations of such a framework. I feel it would benefit greatly from a better differentiation from other frameworks and discuss what the framework should do, and what it should *not* do.

# Comments

Thank you for submitting this paper, which I believe highlights an important need for more thorough versioning and standardization.

However, I believe that in the current state, this paper is rather shallow and I feel it would benefit from a more clear separation from other frameworks (e.g. for deployment in a large-scale environment) and using that space for more detailed insights into the compared tools.
In particular, I feel that many of the points raised in 5.1 should already be addressed in this work.

Secondly, I believe many points raised in the research agenda are active research fields, and a more insightful comparison would be beneficial to clarify the position of this framework in the larger context of ML research.

## Details

3.2: What are the benefits for versioning/automation of such tools compared to simply git+CI/CD? Can you provide a more detailed description of what these tools *do to help?*

4.2, Observation 3 (automatic dependency discovery): I am unconvinced of the actual benefits of discovering software dependencies compared to using docker images; I rather believe that using docker images is a strictly more powerful approach. Docker separates the dependencies of the host system from the container (all we need to know from the host are the kernel and docker version), reducing interference. Furthermore, only capturing libraries and version numbers often is not enough: What if libraries need to be configured properly? Or do not work if another library is installed? A docker image contains all these interactions in an isolated environment that can also be versioned easily (as a simple dockerfile specifies the image).

5.1, (user experience): I believe addressing the user experience (ease of version and automation) of each tool in this work would make this framework much more valuable, and I feel it should be addressed in this paper already.

5.1, (performance & benchmarking): It seems to me that the performance overhead of a (useful) versioning framework should be negligible. The main requirements for computational resources (training, eval, ...) seem orthogonal to how exactly the artifacts are versioned. Maybe it would make sense to plot the framework overhead for different artifact counts and sizes, but this could already be addressed by this paper without a user study and using random artifacts.

5.2: I question whether a single framework should attempt to capture everything. Especially the development and deployment stages seem to be very environment-specific, e.g. which training or monitoring capabilities are available; this all seems very different from code versioning and the current text does not convince me of the benefits of packing all into a single framework. The line "if and how a distributed system can be retrieved and set up automatically" best highlights this problem for me. This is an incredibly large area in itself. Maybe you should focus more on how the framework could *connect* to existing systems for large-scale deployment, instead of trying to try and capture it all under the umbrella of this framework.

5.3. (reasoning): This seems to boil down to "good coding practices", i.e. explaining your work instead of just throwing artifacts out there. This is already possible when versioning code (and even data), as you describe yourself in the comparison between snapshots and versioning. To me, this seems like an appeal to the developer(s), not a missing capability of the framework. Moreover, is this kind of reasoning enough? There is a lot of research in the field of "interpretable ML" which could be incredibly important to truly understanding why something can fail even if all commands run through.

5.3. (sensitivity): Understanding the sensitivity of ML to inputs and hyperparameters is a research discipline on its own and goes far beyond "value ranges" for e.g. hyperparameters. For example, consider the research into Bayesian optimization for hyperparameters. The "while not straightforward" does not appreciate the complexity of this problem at all and I feel that it mischaracterizes the difficulty of adding such capabilities, which may be borderline impossible in a general way.

## Nitpicks

- The inclusion of TensorBoard in the framework seems rather out-of-place. It is the only example for the category "visualization" and seems in general to have a different focus than the other tools which makes me question whether it would make more sense to exclude it.
- Why introduce the Development and Operation stages in detail if they are not relevant for the framework? Focusing on the experiment stage might help to make the paper more on-point.
- I believe it would help the reader to put a legend for the notation in the table captions, it is currently rather hard to jump back and forth between the tables with results and the text explaining the notation.
- I do not understand the arrows in figure 3: why are code and outputs not published?

**Expertise:**

Actively publishing in this area

**Useful:**

yes

---

### Official Review · Reviewer_zoNc · 2021-12-06
**Paper Review**

**Decision:**

Weak accept: good paper with flaws that can be fixed in three months

**Review:**

## Summary

The authors have reasonably defined reproducibility, identified artifacts that during ML lifecycle that directly affect reproducibility, and have proposed a framework for categorizing system support for reproducibility. Using this framework, the authors presented a systematic survey on the support status of mainstream tools, and from which they found generally good reproducibility support with rooms to improve, including hardware dependency tracking and interoperability.

## Discussions

While I think the authors did a reasonably comprehensive job in covering the major aspects of reproducibility, I would like to add a few points for consideration/discussion.

1. Model throughput reproducibility? In production settings we look at two major metrics: accuracy and throughput, and the later is needed for business purposes, for example: we need this inference to be done within 100ms before the user gets impatient. Thus, during the design stage we expect the model to hit XYZ queries/s, we would like it to hit a similar QPS of XYZ in production setting, and it will be highly desirable to capture the runtime performance aspect of ML training;
2. Run to run variance? in ML we often tolerate a small run to run variance, and I think it would be important to explicitly consider this, as the strictest sense of reproducibility does not and need not apply. This variance can stem from random initialization, random optimizer walk, asynchronous updates, and the order with which the data is fed. How would a perfect tool accommodate to this?

## Nits

- 2.1: white space after dot
- 3.1 explicitely -> explicitly
- 4.2 e.g.,, -> e.g.,
- 4.2 One one end -> on one end

**Expertise:**

Follow the literature closely, last published 5+ years ago

**Useful:**

yes

---

### Official Review · Reviewer_R7hz · 2021-12-12
**Review of "ML Reproducibility Systems"**

**Decision:**

Weak reject: interesting papers with flaws, not sure if they can be fixed in three months

**Review:**

Summary


The paper proposes a framework for evaluating the reproducibility capabilities of ML tools, applies the framework to 12 popular ML lifecycle management tools, and provides a set of observations and research challenges.


The paper is well-written and easy to follow. It has the potential to be a valuable paper. However, it needs a significant amount of additional work.


Here are my comments and suggestions on how to improve the paper.


Comments


ML Project Artifacts


* Output data and metrics - The point of reproducibility research space is that these outputs can be recreated. These artifacts are not “essential for achieving reproducibility”, but can be considered essential for “verifying reproducibility”. Even, reproducibility can be verified with just a fraction of those outputs (in case they are large in volume). If the focus on the paper is to “achieve reproducibility” then these two should be excluded from the artifacts, however, if the focus is to “verify or evaluate reproducibility” these two can remain in the list -- but more information needs to be added in case these are large in size.


* Software dependencies - these dependencies should include software itself and necessary packages (e.g., Python and NumPy). In other words, add that dependencies include the software itself, which is now missing.


* Experiment - Consider renaming this to “Computational workflow”, and update the language in the paragraph.


* Don’t use “achieve perfect reproducibility” as reproducibility is already binary (either something is reproducible or it is not). Change this to “to enable reproducibility” or “to verify reproducibility” or “evaluate reproducibility”. If you’d like to introduce nuance in reproducibility, you should add a definition on partial reproducibility and a reproducibility range in 2.2.


Table 1: add what each letter means in the table caption


Figure 2. Creating the experiment and reproducing the experiment parts should be divided.
Also it seems that both versions (light blue and light yellow) are fetched at the same time, is this intentional? It is unclear what “automation” is in this figure.


The Systems
* It would be helpful for the reader to see a bullet list of each of the projects with a 1-2 sentence description (as many will not be familiar with at least some of the tools). Can these systems be used with each other or they serve the same purpose?


* Table 3 is the essence of the paper, yet it is not very informative. I suggest sorting the tools from best to worst, adding a rating based on its popularity (ie GitHub stars, or no of contributors), maybe a comment, and setting it to landscape. That way a reader can immediately see which tool has the best reproducibility support.


* The observations are somewhat in contradiction to the results presented in the table.


* Observation 1 is a bit arbitrary and vague (ie, all systems have some versioning). I suggest removing it and instead adding a project description for each project and a conclusion pointing out which systems are best for reproducibility.


* Observation 2 - do the tools have integration/support for git? If so, they don’t need to ‘reinvent the wheel’ and implement their own versioning. Also I wouldn’t say that full versioning is rare given that there are many “Vs” in Tab 3


* Observations 3 and 4 are critical. Putting higher emphasis on them would be good and also adding in Tab 3 if the tools support dependency capture, containers, VMs. It is also important to note that Python based platforms can use built in dependency capture. Would that be possible for the rest of the platforms?


* Observation 5 is confusing. Continuous integration (CI) allows testing a ML workflow on various software versions and systems seamlessly. I expect any of these systems can be tested with CI, so I don’t agree with this observation. Also, I don’t understand why would one want to “export experiments to a different environment”, when the original one can be exported or encapsulated into a virtual container. Also, “different types of reproducibility” sounds very odd.


* Observation 6 is vague (“similar but different” sounds unscientific), and it should be removed. The first sentence is in contradiction with Observation 1. The main points from this observation can be added to the description of each tool.


* Observation 7 is essentially saying that the presented methodology is bad. Some of the points here may be true, but the fact that “classification is difficult and messy at times” just showcases the flaws in the methodology. If the paper is to be resubmitted, this observation should be removed and difficulties of classification should be discussion elsewhere.


Research Agenda


* The Framework extensions section is not very convincing.


* The “Entire Lifecycle” points to a major flaw of the methodology, which is that the paper only evaluates the “experimentation phase of the ML lifecycle”. I don’t think that is the case. First, I don’t see why the same observations and reproducibility metrics cannot be applied to the development phase of the ML process. Second, I don’t understand why one would measure reproducibility of a model in the operation phase unless that model is not updated in real time. This paper uses “reproducibility” where it should just use code/model “testing”. Further, does any of the systems support/encourage/enforce code testing?


* The “Scalability” subsection doesn’t actually speak about the system scalability, but stand-alone parts of the ML lifecycle that are not incorporated as part of the main flow.


* “Reasoning” is not very convincing. Surely all systems support process/model/code documentation?


* “Sensitivity” has a misleading name, and it points to the bad methodology (ie “we do not yet understand how important each of the artifacts is individually”).


Minor:
* There is no space after a full stop and a capital letter in several places, ie, 2.1 first paragraph “.E”


* python -> Python with capital P

**Expertise:**

Actively publishing in this area

**Useful:**

yes

---

### Official Review · Reviewer_Jk4v · 2021-12-18
**An important topic worthy of attention. Is it within scope of systems?**

**Decision:**

Weak accept: good paper with flaws that can be fixed in three months

**Review:**

This paper has the noble aim of  understanding the requirements of reproducibility for ML applications and to assess the
reproducibility support of existing ML lifecycle management systems. Although the word "systems" is mentioned, its use here strongly suggests software engineering as opposed to traditional computing systems. That editorial concern aside, it is clear that machine learning is becoming an increasing part of what we do in systems research in general, so I will proceed with the assumption of this being in scope.

Overall, I found this paper to be fairly well-written. Throughout the text, however, I feel that more use of active voice would make the paper read a bit stronger. So I would ask the authors to please consider an editorial to make the writing a bit more active--and a bit tighter as well. The abstract, or example, is a bit too long. I also think the "Research Agenda" is a bit inappropriate for a technical paper. This makes the paper conclude more like a proposal when, in fact, there has been substantial scholarly effort leading into this section. Consider significantly shortening this section or removing it, unless it is really contributing to the contributions of the paper.

Regarding the motivation for the paper, there are few points that could be sharpened:
The authors claim (with citation) that "ML model creation is experimentation-based, making it an ad-hoc process with frequent jumps between the different lifecycle phases". I'm not sure this statement is true and certainly will benefit from a clear example/examplar that supports this claim. As someone who works with ML codes (mostly in the Computer Vision space), we see projects that are much more like typical software development lifecycles, since they often result in proper software releases and tools. It might be beneficial for the authors to be more clear about the actual types of ML projects they are considering as part of this study (and which are being left out). Later in the same paragraph, the authors state, "As a result, existing software engineering methodologies and best practices cannot be directly applied." I don't agree with this statement either. We already know that agile methods (outside of the ML space) also jump freely between lifecycle stages, so it would seem at first blush like the same limitations apply to many other types of software projects.

The authors state, "To prevent an impending reproducibility crisis [23, 25, 49], both research and industry have spent significant effort on trying to incorporate reproducibility as a first-class citizen into the lifecycle management systems." Given that one of the projects cited is TensorFlow, the authors may wish to be aware of the TensorFlow Models Garden effort at Google, https://github.com/tensorflow/models and an experience report at https://arxiv.org/abs/2107.00821v1.

I really like the discussion of artifacts in section 3.1. But I found myself wondering, why isn't any technical report or paper describing an ML Project not an artifact, too.  For many projects, we have observed that the paper often amounts to the equivalent of a "requirements document" that describes what the model is supposed to do and how it does it. Could it be that many ML projects amount to being part of a general class of research software that is often linked to a paper or set of papers? We have observed this in many NLP and CV projects.

Overall, I find the study to be well-constructed. However, I did find myself wondering whether the study itself is reproducible. This is an anonymous submission but it would be nice to know more about the methods used and whether any tools/datasets are being made available to the community to reproduce or extend these efforts.  If this paper is accepted, I hope the authors can speak more directly about how reproducible this study actually is and what tools/scripts/datasets will be made available to the community.

In closing, there are some flaws in this paper but most can probably be addressed with routine revisions.  This work is definitely worthy of attention in the software engineering community and, although a bit of a stretch for systems, probably has a home in JSys.




**Expertise:**

Actively publishing in this area

**Useful:**

yes

---

### Meta-Review · Area_Chair_5Kv4 · 2021-12-22

**Recommendation:** Revise
**Confidence:** 5

**Metareview:**

Dear authors,

Thank you for submitting to JSys!

All reviewers concur that this is a valuable piece of work, but it still requires some work before publication. Here is a summary of the main points that should be addressed in your revision.

- The purpose of a SoK paper is to bring order and clarity within a given (sub)field of research. This paper itself acknowledges the classification enabled by the framework is still "messy", which shows that the proposed classification framework still needs improvements. In particular, while Table 3 is the central piece of the paper, it does not allow for easy parsing of the information.
	- Consider adding some color-coding for a more visual output (e.g., N -> \colorbox{red}{N})
	- Consider highlighting the "best" tools for reproducibility and sort them accordingly
	- Consider adding more context about the tools, e.g., their popularity (#stars on GitHub) or their liveness (update frequency and/or number of contributors).
	- Consider splitting the table in two sub-tables, one for "versioning" and one for "automation." This may facilitate ordering of the systems to better highlight which is "best."
	- Please repeat the meaning of the letters in the caption.
- The analysis/description of the 12 systems is very limited. Please add a short description of each (possibly at the end of Sec.4.1) such as to guide the unfamiliar readers looking for the best tool for their next project. Please also explain why a certain system "ticks the boxes" of your classification.
- What about a classic "git + CI/CD" pipeline? Please add this to your list of "systems" to compare it to the others, which would provide an interesting baseline to compare with (and motivate the interest of) the specialized tools presented in the paper.
- Several reviewers made specific comments about the ML artifact classification (Sec.3.1) and analysis observations (4.2). Please address them.
- The "research agenda" section appears a bit simplistic. The reviews contain detailed points that are unclear or too abstract. Please address them.
- Highlight more clearly earlier on in the paper what the scope of the current framework is. If you intend to focus on the experimental phase, then make it clear. This may allow sparing the description of the development and operational phases and make the paper more to the point.
- Please add a discussion on how the systems analyzed support ML performance analysis, such as:
	- How to account for run-to-run variance?
	- How to measure and replicate the model inference throughput?
- Please comment on the replicability of that study itself. Are there any tool/dataset that you plan on releasing? The paper mentioned more details in "supplementary material" but without any link to it!
- In the case of (very) large number of outputs; is having all output data really a requirement in order to "verify reproducibility"? Please comment.
- Clarify whether you consider reproducibility to be binary or not; In the latter case, explain how "partial reproducibility" could be defined.
- The reviews contain many other secondary points and suggestions for improvements, please consider them.
- Consider using the "plainurl" bibtex style and provide an URL or DOI for each of the references in your bibliography.

The complete reviews will be released shortly (it is delayed for technical reasons... sorry for that!).

We look forward to your revision!

---

### Decision · Program_Chairs · 2021-12-20

**Decision:**

Revise

**Comment:**

Dear authors,

All reviewers agree on the value and interest of this work, but also that it still requires significant work to be considered for publication. We would like to invite you to **revise and resubmit** your manuscript.

The area chair will finalize the meta-review in the next few days, which will outline the points where updates are expected. As per JSys policies, you will have 3 months from the date of the meta-review in order to revise and resubmit your manuscript.

From https://www.jsys.org/cfp/#revisions-and-review-process
===
The revised manuscript is submitted using the same OpenReview forum used for the initial submission; use the "Revision" button to submit the revision, which must contain two parts:
- A revision plan explaining the changes made, and how they address reviewer concerns;
- The revised manuscript, with changes highlighted for easy reviewing.
===